# Assessing Support for Advantaged and Disadvantaged Groups: A Comparison of Urban Food Environments

**DOI:** 10.3390/ijerph16071135

**Published:** 2019-03-29

**Authors:** Ryan Storr, Julia Carins, Sharyn Rundle-Thiele

**Affiliations:** 1Social Marketing at Griffith, Griffith Business School, Griffith University, Nathan, QLD 4111, Australia; j.carins@griffith.edu.au (J.C.); s.rundle-thiele@griffith.edu.au (S.R.-T.); 2Defence Science & Technology Group, Land Division, Scottsdale, TAS 7260, Australia

**Keywords:** food environment, public health, obesity, socio-economic status, measurement

## Abstract

Individuals from lower-socio-economic status (SES) communities have increased risk of developing obesity in developed countries such as Australia. Given the influence of the environment on dietary behaviour, this paper seeks to examine food environments in areas of differing social advantage. An established measurement tool (the NEMS—Nutrition Environment Measurement Survey), that captures aspects of support for healthy eating within restaurants (NEMS-R) and grocery/convenience stores (NEMS-S), was applied to both a high-SES and a low-SES suburb within Brisbane, Australia. The study found a significantly more supportive restaurant food environment in the high-SES suburb, with greater access to and availability of healthful foods, as well as facilitators for, reduced barriers to, and substantially more nutrition information for healthful eating. A higher number of outlets were found in the high-SES suburb, and later opening times were also observed. Overall, the results from stores (NEMS-S) suggest poor support for healthful eating across both suburbs. This study highlights how food environments in low-SES regions continue to be less supportive of healthful eating. Public health strategies must move beyond individual-focused strategies to ensure that our most disadvantaged, low-SES communities have an equal opportunity to access healthful foods.

## 1. Introduction

Obesity is a global epidemic [1]. Australia is the fourth heaviest country per capita in the world, with more than 25% of the adult population considered obese [2], costing the Australian economy $56.6 billion each year [3]. Poor dietary choices contribute to the epidemic [4]. Furthermore, studies highlight differential access to healthful food, noting that the food environment surrounding individuals is a major factor in poor diet and obesity [5,6]. Disparities have been observed between advantaged and disadvantaged individuals with regard to diet quality [7,8,9,10,11]. Consensus has emerged showing that higher dietary scores (indicating more healthful diets) are associated with advantaged individuals, communities, or both; furthermore, higher income and higher educational attainment have previously been associated with healthier eating [8,9,12]. Socio-economic status (SES), which encompasses income, educational attainment, unemployment, and dwellings with or without motor vehicles, has been operationalised as the Socio-Economic Index for Areas (SEIFA) [13]. Research suggests that disadvantaged communities have the highest obesity prevalence rates and are the most reliant on public health services [14]. Studies have found that SES is related to diet, for better or worse [10,14,15,16,17,18]. Given the influence of the environment on dietary behaviour, it is important to examine the environmental opportunities (or lack thereof) for healthful eating in low- and high-SES communities in Australia.

The food environment refers to the places individuals go to purchase and/or consume food and beverages. The consumer food environment (or consumer nutrition environment) considers what individuals encounter within a food outlet [19], and includes accessibility, availability, information, facilitators and barriers, and pricing support [17,20]. Inequalities have previously been identified. For instance, ‘food deserts’ (areas with poor access to healthful foods) appear more prominent in low-SES communities, with strong evidence of this effect in the United States, but less convincing evidence in Australia [21]. Some Australian studies have found reduced access to healthful foods in low-SES suburbs using supermarkets and convenience stores as proxies for healthful food stores and takeaway outlets as proxies for unhealthful food providers [16,22,23,24], without considering the consumer environment within these businesses. Few studies in Australia have examined the consumer food environment, within stores or outlets, to determine if consumers from different areas experience differences in support for healthful eating when purchasing food. 

Reduced availability of healthful food items has been noted in outlets from low-SES communities, with healthful items given less shelf space and restaurant menus containing fewer healthful options than in high-SES communities [12,17,25]. In Australia, the availability of healthful food within stores has been observed to be lower in low-SES areas [24,26,27], although in other studies it was only marginally so [22], or even similar [28]. The shelf space dedicated to non-core (less healthful) items was also found to be similar in low- and high-SES suburbs [29]. Research examining the other aspects of the consumer nutrition environment within Australia is less common, and findings have been inconsistent. The quality of healthful foods has been found to be comparable in some parts of Australia [28], but substantially higher in advantaged areas elsewhere [26]. The price of a healthy basket of food has been found to cost more in high-SES areas in some studies [22,26,27], whereas it has been comparable in other studies [28]. These studies have examined ‘absolute’ price—comparisons have not been made regarding pricing support (e.g., between low and high fat alternatives or between white and wholemeal bread to determine whether pricing encourages the purchase of the healthier alternative). Most Australian states now require the provision of nutrition information in the form of kilojoule labelling in some retail food outlets (RFO). Only one study, conducted prior to the introduction of the legislation, has examined the provision of nutrition information, finding that more information was provided in low- and medium-SES areas than in high-SES areas [30]. 

Researchers have called for further investigation to enhance the understanding of the food environment and its relation to socio-economic status [24,31]. Most studies that have examined the interplay between SES and the consumer food environment have been conducted in the Americas, primarily the United States [7,8,9,32]. Taken together, there is a growing need to extend our understanding of how food stores and outlets support healthful food choices across the population range. Given that gaps exist with respect to key aspects of environmental support for healthful eating, this study sought to apply an established food environment measure to both a high-SES and a low-SES community in Australia to quantify aspects of support for healthful eating.

## 2. Materials and Methods

The food environment has been measured in different ways, with the Nutrition Environment Measures Survey (NEMS), one of the most common instruments [33], known for its validity, reliability, and reproducibility [34]. The NEMS is based on established dietary guidelines and epidemiological data, and moves beyond assessing availability—incorporating measures of price, quality, information, and promotion [35]. This study followed procedures detailed in the original NEMS studies [17,20] and the NEMS training manual protocols (www.med.upenn.edu/nems). 

The SEIFA [13] was used to select one low-SES (first quintile) and one high-SES (last quintile) suburb in Brisbane, Australia with a sufficient number of food outlets. The high-SES suburb had a population of ~8000 and a SEIFA score of 1124 (quintile 1 range = 1053–1182); the low-SES suburb had a population of ~13,800 and a SEIFA score of 765 (quintile 5 range = 554–936) [13]. The survey area was set around the centre of each suburb (e.g., post office) with a 2 km radius in line with NEMS protocols. Other Australian studies have used radiuses of 800 m [36] and 2.5 km [23,37]. The radiuses were plotted using geographical information system (GIS) software, and the population of outlets was compiled [38] by physically traveling the area. Sample size calculations indicated that 25 outlets were needed in each group (low- and high-SES) to detect a significant (*p* = 0.05) difference in a NEMS score of 5 points (80% power; SD = 6.2 points based on a previous Australian study [39]). 

To assess the food environment, the NEMS-R (for restaurants/takeaway outlets) and the NEMS-S (for grocery/convenience stores) instruments were used, as adapted for Australian metric measurements, common terms, and brands [39]. Assessments were completed by one author, with a 10% test–retest and inter-rater reliability check [17]. NEMS scores and sub-scores were calculated using NEMS protocols, and checked by a second author. Other variables included highest and lowest price; opening, closing, and meal times (breakfast, lunch, and dinner); whether the outlets were in a complex (mall) or stood alone; quality; and direct price comparison of fruits and vegetables. 

Statistical analyses were performed using SPSS (IBM, Armonk, NY, USA). Inter-rater and test–retest reliability was determined using kappa coefficients (κ) and percent agreement. Independent samples t-tests examined differences between suburbs for continuous variables, and chi-square tests and Fisher’s exact test were used for dichotomous variables.

## 3. Results

Enumeration was conducted in May 2016, with totals of 47 outlets (low-SES) and 83 outlets (high-SES) identified in each suburb. Random samples of 30 eating outlets (low-SES) and 35 eating outlets (high-SES) were drawn from the list enumerated in each suburb, providing a sufficient sample size for measurement with NEMS-R, with a buffer. A loss rate of approximately 10% had been identified in previous studies (through closure, rejection, or insufficient details to allow measurement) [17,20,39]. Due to the small number identified, all stores were included in the sample list for measurement with NEMS-S (19 outlets in the low-SES suburb and 17 stores in the high-SES suburb). Assessments were carried out in August 2016 (NEMS-R) and late September 2017 (NEMS-S). Of the outlets sampled, 25 (83%) were assessed with NEMS-R in the low-SES suburb (three restaurant closures, two restaurant rejections), and 29 (83%) were assessed in the high-SES suburb (three restaurant closures, three restaurant rejections). In the low-SES suburb, 16 (64%) eating outlets were inside a complex (shopping mall), compared to only three (10%) in the high-SES suburb. For stores, six (32%) were assessed with NEMS-S in the low-SES suburb (three store rejections, five abandoned during measurement due to cultural diversity of the stock, and five considered unsuitable to commence measurement due to cultural diversity of the stock), and 13 (76%) were assessed in the high-SES suburb (three restaurant closures, three restaurant rejections). The reliability of measurement within this study was high, the percentage agreement across NEMS-R questions was high (mean = 89.92% for inter-rater; 96.75% for intra-rater), and a moderate-high level of agreement was observed according to kappa scores (mean κ = 0.69 for inter-rater; κ = 0.91 for intra-rater). 

### 3.1. Support for Healthful Eating—Restaurants, Cafes, and Other Eating Outlets

The suburbs assessed in this study provided low to moderate support for healthful eating within eating outlets (restaurants/fast food/cafes). NEMS-R scores can range from −24 to +63 for restaurants, meaning that the mean NEMS-R score for the low-SES suburb (*x* = 7.08) was below mid-range, and the mean NEMS-R score for the high-SES suburb (*x* = 20.38) was mid-range (see Table 1). The mean NEMS-R score and all sub-component mean scores (except pricing support) were significantly lower in the low-SES suburb, denoting less support for healthful eating in this suburb. There was no difference in pricing support, indicating that pricing strategies for healthful options and non-healthful options were similar in both suburbs. A difference was found between suburbs for the cost of the cheapest meals, but not for the most expensive meals. Mean scores and suburb comparisons for the NEMS-R key elements are shown in Table 1 below. 

Taken together, the results from the restaurants indicate more support for healthful eating in the high-SES suburb, including greater availability, more information provision, as well as more facilitators and fewer barriers. Pricing support did not differ between the two suburbs, meaning prices were similar when comparing healthful and less healthful alternatives within product categories. Cheaper meal options were available in the low-SES suburb. A number of healthful items were more commonly offered on menus in the high-SES suburb including wholemeal bread, fruit juice, low-fat milk, and non-fried vegetables. However, every restaurant in both suburbs sold diet beverages, while no restaurant offered an all-you-can-eat option or unlimited trips (an enabler of over-eating, therefore a barrier to healthful eating). No restaurant offered smaller portions at a reduced price (a pricing strategy for healthful eating).

Trading patterns were examined to determine whether accessibility differed between suburbs. There was no significant difference between the suburbs in whether outlets traded on each of the week days or weekend days, nor in the frequency of opening for the lunch time meal. However, a higher proportion of outlets were open for breakfast in the low-SES suburb compared to the high-SES on weekdays and weekends. The opposite was observed for the dinner meal, with more outlets open in the high-SES suburb. Closing time was later in the high-SES suburb, although more eating outlets were located in a shopping complex in the low-SES suburb, which may have constrained closing times (low-SES: 21/25 RFOs vs. high-SES: 3/29 RFOs, χ^2^ = 29.5, *p* < 0.001).

### 3.2. Support for Healthful Eating—Convenience and Grocery Stores

Both suburbs provided a moderate amount of support for healthful eating. NEMS-S scores can range from −9 to +52 for restaurants, so the mean NEMS-S score for both suburbs was mid-range, or just below (low-SES suburb *x* = 21.17; high-SES suburb *x* = 15.69). There was no difference between suburbs for the NEMS-S score or the sub-scores of availability and quality. However, the small sample size, and therefore the reduced power to detect a difference, must be acknowledged. Calculations indicate that a minimum sample size of 39 per group provides 80% power at α = 0.05 level to detect a mean difference of 5.5 (the difference we observed) with a standard deviation of 9.6 (the standard deviations that we observed), and our study did not achieve that sample size. There was one significant difference (*p* = 0.035) in the NEM-S pricing support dimension; the lower score in the high-SES suburb indicates less support, or a greater price difference between healthful and non-healthful alternatives in stores. Fruits and vegetables are not included in the pricing support comparison, as they are all considered healthful items, so a comparison of absolute price of these commodities was conducted. Not all stores stocked each variety, so only the most frequently found items were compared, and again the small sample size and reduced power to detect a difference raise concerns. Mean scores and suburb comparisons for the NEMS-S key elements are shown in Table 2.

Overall, the results from stores suggest poor support for healthful eating across both suburbs. The mean NEMS-S score was higher in the low-SES suburb, and the magnitude of the difference between suburbs was similar to that observed in the original US study [17], suggesting that more support may have been present in the stores of the low-SES suburb. However, a statistical difference was not observed (most likely due to inadequate statistical power), and this difference should be explored in larger studies in the future. The low-SES suburb offered more pricing support—in other words, healthful items were priced similarly to their less healthful equivalents, whereas in the high-SES suburb there was less pricing support (a greater price differential).

## 4. Discussion

The results of this study indicate that the high-SES suburb provided more support for healthful eating compared to the low-SES suburb in the prepared food market. Enumeration revealed that the high-SES suburb had almost double the amount of food outlets compared to the low-SES area, despite having a considerably lower population density. This, combined with significantly later closing times found in the high-SES suburb, indicates a higher level of access to prepared food options in the high-SES area. Given that this access co-occurred with higher levels of availability of healthful foods, a greater level of information on healthful eating, and more facilitators and fewer barriers to healthful eating, the food environment in the high-SES offered more support for healthy eating to individuals dining in that area. Another Australian study found a correlation between ‘healthier’ food environments and resulting healthier eating patterns within local populations [24]. This emphasises the potential for flow on effects from the healthfulness of the food environment to the individual, and the implications for low-SES communities which may arise from these combined findings.

Findings of the current study partially align with prior research that has shown differences in the consumer nutrition environment [18,24,26,27]. Critically, these studies focus on one or two elements of the food environment (price and availability) [26,27]; furthermore, one Australian study neglected both restaurants and corner stores [27], and as such they have not measured the food environment as comprehensively as the current study which utilises the multi-dimensional NEMS measures [17,20]. Taken together, all studies highlight socio-economic disparities within the food environment and the tools utilised highlight calls for action to ensure that food environments support healthy food choices. Expanding the focus into restaurant assessment, this study, in line with past research in low-SES suburbs (in Australia and the US), continued to identify lower levels of support for healthy eating in Australia [7,9,18,24,26,27,32]. 

The current study was unable to establish significant differences between low- and high-SES suburbs in relation to stores. This comes in stark contrast to the results identified by the NEMS-R. Higher (but not significant) scores were identified in the low-SES suburb for its total score and all sub-components. A larger sample is needed to confirm these findings, which contradict those of other studies that have consistently associated inferior food store environments with low-SES communities [8,12,24,40,41]. One exception was identified—research conducted in New York found no significant difference in the advertising of healthful vs. unhealthful food between low- and high-SES suburbs [42]. The lack of significance in the current study is likely due to an insufficient sample size. The smaller sample size obtained was caused by the cultural diversity of food offerings in stores, which limited the assessment of some outlets within the low-SES suburb. The Western-oriented NEMS-S tool was not able to assess the food environment within stores supplying Eastern foods. 

This study contributes to the literature with an examination of the consumer environment within outlets. Few Australian studies have explored this; most have focused on the number of (un)healthful outlets, and have instead determined the healthfulness of a community based on the types of RFOs located in a suburb [16,40]. The study findings suggest that Australia appears to display similar nutritional disparities as the United States in relation to SES, as demonstrated in US NEMS studies [7,9,32]. These US studies have also shown lower availability, less nutrition information, and fewer facilitators (or more barriers) in the prepared food market [7,9,14,41]. However, no difference was observed in pricing support for healthful eating between suburbs in prepared food outlets, consistent with the results of other food environment studies identified in a systematic review of the literature [31].

While health education is critical, and has been shown to improve dietary behaviour [43], these findings indicate that even if an individual were knowledgeable about nutrition and possessed motivation to follow a healthful diet, there are many obstacles preventing them from reaching their goal, given the lack of support within the food environment. Considering the significant impact the environment has on individual dietary choices [24,44], the food environment needs to be addressed to enable or amplify other behavioural change efforts, in both low- and high-SES environments [45,46]. Attempts to alter individual behaviour when there is a clear absence of healthful options and information will limit intervention effectiveness. Engaging food outlets to co-create a healthful eating environment is recommended [42]. 

Of concern, low-SES consumers are frequently dependent on the food environment that surrounds them, with many of these communities lacking transport and transport infrastructure [18,24,47,48]. A food environment that does not support healthful eating in a low-SES suburb poses a considerable health risk to that community, given that the dependence on outlets available in the local area is highest for people living in low-SES areas [14,18], and links between the environment and body-mass index (BMI) are clear [15,41,49]. This has implications for the community itself, and the broader Australian society, through the increased financial burden of ill-health caused by poor diet [3]. Limited access to outlets providing healthful foods creates further disparities, further extending the social divide [12,50]. This study portrays a relatively unsupportive food environment in one low-SES area in Australia, supporting earlier research [14,24,51]. This is a significant, avoidable economic cost [3]. Strategies to address unsupportive food environments, including regulation or stronger self-governance of food outlets, are needed to focus change efforts on food provision in the food supply chain [47]. The provision of a greater availability of healthful alternatives and improving accessibility to healthful food outlets may help achieve this [7,8]. 

This study measured support for healthful eating in the food environment, delivering evidence of low support for healthful eating. Further research is recommended to increase sample sizes across low-SES areas to ensure that definitive conclusions can be drawn from the findings obtained in the current study. This study acknowledges that other factors are associated with poor diet. In line with a call for a greater scope of analysis, future research is needed to develop tools that assess the supportiveness of the Australian food environment, regardless of ethnic diversity within those regions [24]. Future research could pair observational assessments of the food environment with research seeking to understand the perceptions of consumers and retailers in those environments. Qualitative research into the reasons why individuals in low-SES Australian suburbs consume the foods that they do would provide a valuable, in-depth understanding of the phenomena noticed in this study, and would explore how food environments are shaped by the demands of the consumers within them, and in turn how consumer demand is shaped by the environment. 

A study that aims to track the food environment over a period of time is recommended, and is in line with the call for a wider research focus [52]. Specifically, measurement of food environments over repeated time periods is recommended to focus retailers’ attention on food healthfulness. Combined measures (audits, surveys, focus group(s), and/or interviews and BMI measures) would facilitate an understanding of how food environment change can deliver changes benefitting individuals in low-SES areas.

## 5. Conclusions

This study explored differences between low- and high-SES suburbs in terms of support for healthful eating in retail food outlets that provide prepared meals, finding a greater level of support in the high-SES area. The tools used to assess these outlets were adequate across ethnically diverse environments; however, the equivalent tool for stores was not. Given the cultural diversity noticed in Australian communities, particularly low-SES communities, additional tools are needed. These findings have numerous implications for health in these communities and more broadly for society, and support a growing body of literature suggesting that healthful eating requires a supportive food environment.

## Figures and Tables

**Table 1 ijerph-16-01135-t001:** Indicators of support for healthful eating by suburb socio-economic status (SES) for eating outlets.

	Low-SES (n = 25)	High-SES (n = 29)		
	Mean	Mean	t	*p*
**NEMS-R Score**	7.08	20.38	3.768	<0.001
Information	1.16	5.83	3.013	0.004
Availability	7.00	13.93	4.757	<0.001
Facilitators/barriers	0.24	2.17	2.168	0.035
Pricing support	−1.32	−1.55	0.489	0.627
**Absolute Price**	**Mean**	**Mean**	**t**	***p***
Cheapest meal	$6.11	$8.30	2.406	0.020
Most expensive meal	$17.48	$20.79	1.116	0.270
**Healthful foods present**	**Count**	**Count**	**χ^2^**	***p***
Wholemeal bread	3/25	19/29	15.927	<0.001
Fruit juice	5/25	22/29	16.759	<0.001
Low fat milk	9/25	25/29	14.513	<0.001
Non-fried vegetables	5/25	18/29	9.718	0.002
**Accessibility**	**Count**	**Count**	**χ^2^**	***p***
Thursday breakfast	21/25	12/29	10.262	0.001
Friday breakfast	21/25	12/29	10.262	0.001
Saturday breakfast	21/25	11/29	11.803	0.001
Sunday breakfast	19/25	9/29	10.873	0.001
Thursday dinner	12/25	23/29	5.771	0.016
Friday dinner	12/25	24/29	7.299	0.007
Saturday dinner	11/25	24/29	8.844	0.003
Sunday dinner	11/25	23/29	7.178	0.007
**Closing time**	**Mean**	**Mean**	**t**	***p***
Thursday closing	18:25	20:16	2.987	0.004
Friday closing	18:24	20:33	3.768	<0.001
Saturday closing	18:00	20:30	4.040	<0.001
Friday closing	18:17	20:12	2.869	0.006

**Table 2 ijerph-16-01135-t002:** Indicators of support for healthful eating by suburb SES for stores.

	Low-SES (n = 6)	High-SES (n = 13)		
	Mean	Mean	T	*p*
**NEMS-S Score**	21.17	15.69	1.161	0.262
Availability	16.00	14.38	0.531	0.602
Price	1.67	-0.69	2.289	0.035
Quality	3.50	2.00	1.077	0.297
**Fruit and vegetable pricing**				
Bananas	3.06 (n = 5)	4.43 (n = 10)	1.554	0.144
Apples	3.97 (n = 5)	4.69 (n = 10)	0.825	0.424
Oranges	2.71 (n = 3)	4.08 (n = 8)	1.612	0.141
Carrots	1.32 (n = 4)	2.40 (n = 7)	4.020	0.004
Tomato	4.25 (n = 4)	5.28 (n = 9)	0.796	0.443
Lettuce	2.37 (n = 4)	3.45 (n = 8)	1.420	0.186

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
