# Peer review of "Assessing Support for Advantaged and Disadvantaged Groups: A Comparison of Urban Food Environments"

_ijerph, 2019, doi:10.3390/ijerph16071135_

Round 1
Reviewer 1 Report
1) Overall a great paper!
2) I wish you put some of the data (mean scores) in the text -- you discussed the scores (lines 132-134) without telling us the mean scores or even referring to table 1. If I had the paper in my hand and could 'glance' to the right to see the table, I suppose that would be fine -- but on a computer, I had to scroll to find the data.
3) I find it hard to read your table -- I assume you mean "Absolute Price" and "Healthful Foods Presented" to be headers -- is it possible to put a space after "pricing support" and before "absolute price" to make it easier to "see" the breaks you want us to find in the tables?
4) I am not finding enough discussion about the LACK of statistical significance in the grocery stores between low and high SES. Its as if you wanted to brush it under the rug. I find it interesting that the Low SES had a numerically higher (though non-significant) NEMS score. Can't you calculate your power post-hoc so you can attempt to explain this observation? I do appreciate the fact that the instrument may not be appropriate culturally....
5) Overall, your team writes well! This is a great project.
Author Response
Response to reviewer comments: ijerph-468844 "Assessing the great Australian divide: A comparison of Food Environments"
We wish to extend our thanks to the editors and reviewers for their insightful remarks, and for helpful suggestions on how we might improve the paper. We feel that the paper is much stronger following our revision based on the comments provided by the editors and reviewers. We outline each of the reviewer’s comments and our responses below.
Reviewer: 1
1. I wish you put some of the data (mean scores) in the text -- you discussed the scores (lines 132-134) without telling us the mean scores or even referring to table 1. If I had the paper in my hand and could 'glance' to the right to see the table, I suppose that would be fine -- but on a computer, I had to scroll to find the data.
Mean scores have been included in lines 133-134 as requested. We now refer to mean scores presented in table 1 as follows:
NEMS-R scores can range from -24 to +63 for restaurants, meaning the mean NEMS-R score for the low-SES suburb (x=7.08) was below mid-range, and the mean NEMS-R score for the high-SES suburb (x=20.38) was mid-range (see table 1).
2. I find it hard to read your table -- I assume you mean "Absolute Price" and "Healthful Foods Presented" to be headers -- is it possible to put a space after "pricing support" and before "absolute price" to make it easier to "see" the breaks you want us to find in the tables?
Table 1 has been amended as recommended. Please see the paper for the amended Table 1.
3. I am not finding enough discussion about the LACK of statistical significance in the grocery stores between low and high SES. Its as if you wanted to brush it under the rug. I find it interesting that the Low SES had a numerically higher (though non-significant) NEMS score. Can't you calculate your power post-hoc so you can attempt to explain this observation? I do appreciate the fact that the instrument may not be appropriate culturally....
The lack of statistical significance and numerical difference is now discussed as requested. The sample sizes set for our study were sufficient to provide 80% power. Moving forward larger sample sizes are needed to draw definitive conclusions. The paper now directly reports lack of statistical significance and discusses power as follows:
There was no difference between suburbs for NEMS-S score, or the sub-scores of availability and quality, however the small sample size, and therefore the reduced power to detect a difference, must be acknowledged. Calculations indicate that a minimum sample size of 39 per group provides 80% power at α = 0.05 level to detect a mean difference of 5.5 (the difference we observed) with a standard deviation of 9.6 (the standard deviations that we observed); and our study did not achieve that sample size.
This is discussed in the Discussion section as follows:
The current study was unable to establish significant differences between low and high-SES suburbs in relation to stores, this comes in stark contrast to the results identified from NEMS-R. Higher (but not significant) scores were identified in the low-SES suburb for total score, and all sub-components. A larger sample is needed to confirm these findings, which contradict those of other studies, which have consistently associated inferior food store environments with low-SES communities [8,12,24,41]. One exception was identified, research conducted in New York found no significant difference in advertising of healthful vs unhealthful food between low and high-SES suburbs [47]. The lack of significance in the current study is likely due to an insufficient sample size. The smaller sample size obtained was caused by the cultural diversity of food offerings in stores, which limited the assessment of some outlets within the low-SES suburb. The western oriented NEMS-S tool was not able to assess the food environment within stores supplying Eastern foods.
And the paper acknowledges the sample size limitations as follows:
This study measured support for healthful eating in the food environment delivering evidence of low support for healthful eating. Further research is recommended to increase sample sizes across low-SES attained in the current study to ensure that definitive conclusions can be drawn.

Reviewer 2 Report
The article is about Assessing the relation between the NEMS, availability, quality, price and healthy eating within retail outlets and stores in high-SES and low-SES suburb in Australia. There are some points to be addressed:
- The abstract is just about the study within outlets and didn't say anything about the stores.
- In some parts the references don't folllow the format in the text. For example line 236 and 246 that there is just name and year.
- The author can work on the discussion and conclusion to improve these sections. The discussion should be much more organized and the advantages of this study discussed clearly compared with the previous studies. I didn't get your novel conclusion in this study please clarify.
- I suggest the description of stores in line 255 move to the place that this term was used for the first time in this paper.
- In table one the fraction of sunday dinner in high-SES should be fixed to 23/29.
- I suggest to change your title to the one that is much more appropriate and specific in relation to your paper.
Author Response
Response to reviewer comments: ijerph-468844 "Assessing the great Australian divide: A comparison of Food Environments"
We wish to extend our thanks to the editors and reviewers for their insightful remarks, and for helpful suggestions on how we might improve the paper. We feel that the paper is much stronger following our revision based on the comments provided by the editors and reviewers. We outline each of the reviewer’s comments and our responses below.
Reviewer: 2
1. The abstract is just about the study within outlets and didn't say anything about the stores.
Additional sentences have been added to include elements about stores within the study (lines 20-22), the abstract is now as follows:
“Individuals from lower-socio economic status (SES) communities have increased risk of developing obesity in developed countries such as Australia. Given the influence of the environment on dietary behaviour, this paper seeks to examine food environments in areas of differing social advantage. An established measurement tool (the NEMS—nutrition environment measurement survey), that captures aspects of support for healthy eating within restaurants (NEMS-R) and grocery/convenience stores (NEMS-S), was applied to both a high-SES and low-SES suburb within Brisbane, Australia. The study found a significantly more supportive restaurant food environment in the high-SES suburb, with greater access to and availability of healthful foods, as well as facilitators for, reduced barriers to and substantially more nutrition information for healthful eating. A higher number outlets was found in the high-SES suburb, and later opening times were also observed. Overall, the results from stores (NEMS-S) suggest poor support for healthful eating across both suburbs. This study highlights how food environments in low-SES regions continue to be less supportive of healthful eating. Public health strategies must move beyond-individual focused strategies ensuring our most disadvantaged, low-SES communities have an equal opportunity to access healthful foods.”
2. In some parts the references don't follow the format in the text. For example line 236 and 246 that there is just name and year.
Thank you for this comment. References have been amended for consistency as recommended.
3. The author can work on the discussion and conclusion to improve these sections. The discussion should be much more organized and the advantages of this study discussed clearly compared with the previous studies. I didn't get your novel conclusion in this study please clarify.
The discussion has been examined to more carefully outline the contribution of this paper. For example the paper now states:
Findings of the current study partially align with prior research that has shown differences in the consumer nutrition environment [18,24,26,27]. Critically however, these studies focus on between one-two elements of the food environment (price and availability) [26,27], furthermore, one Australian study neglected both restaurants and corner stores [27], and as such they have not measured the food environment as comprehensively as the current study which utilises the multi-dimensional NEMS measure [17,20]. Taken together, all studies highlight socio-economic disparities within the food environment and tools utilised highlight calls for action to ensure food environments support healthy food choices. This study in line with past research in low-SES identify lower levels of support for healthy eating, in both Australia and the US in low-SES suburbs [7,9,18,24,26,27,32].
4. I suggest the description of stores in line 255 move to the place that this term was used for the first time in this paper.
The description of stores has been moved to the beginning of the study where the term was first used as requested. The paper now states:
To assess the food environment, the NEMS-R (for restaurants/takeaway outlets) and the NEMS-S (for grocery/convenience stores) instruments were used, as adapted for Australian metric measurements, common terms and brands [39].
5. In table one the fraction of Sunday dinner in high-SES should be fixed to 23/29.
Thank you. This typo has been corrected.
6. I suggest to change your title to the one that is much more appropriate and specific in relation to your paper.
The title of the paper has been changed to Assessing support for advantaged and disadvantaged groups: A comparison of Urban Food Environments.
